# Water Consumption during a School Day and Children’s Short-Term Cognitive Performance: The CogniDROP Randomized Intervention Trial

**DOI:** 10.3390/nu12051297

**Published:** 2020-05-02

**Authors:** Alina Drozdowska, Michael Falkenstein, Gernot Jendrusch, Petra Platen, Thomas Luecke, Mathilde Kersting, Kathrin Jansen

**Affiliations:** 1Research Department of Child Nutrition, University Children’s Hospital, Ruhr University Bochum, 44791 Bochum, Germany; luecke.thomas@ruhr-uni-bochum.de (T.L.); mathilde.kersting@ruhr-uni-bochum.de (M.K.); kathrin.jansen@ruhr-uni-bochum.de (K.J.); 2Institute for Work, Learning and Ageing (ALA), 44805 Bochum, Germany; falkenstein@ala-institut.de; 3Department of Sports Medicine and Sports Nutrition, Ruhr University Bochum, 44801 Bochum, Germany; gernot.jendrusch@ruhr-uni-bochum.de (G.J.); petra.platen@ruhr-uni-bochum.de (P.P.)

**Keywords:** cognition, executive function, hydration, school children

## Abstract

There is still little research examining the relationship between water consumption in school and specific cognitive performance. The aim of this cluster-randomized intervention CogniDROP trial was to investigate the short-term effects of drinking water during the morning on executive functions. The participants were from the 5^th^ and 6^th^ grade of a comprehensive school in Germany (14 classes, *n* = 250, 61.6% boys). The classes were randomly divided into an intervention group (an education on healthy drinking behavior and a promotion of water consumption) and a control group. A battery of computerized tasks (Switch Task, 2-Back Task, Corsi Block-Tapping Task and Flanker Task) was used to test executive functions. Urine color and thirst were evaluated to check the hydration level. Physical activity over the past 24 h was measured using GT3X ActiGraph. A non-linear relationship was observed between the amount of drinking water and executive performance. Consuming water up to 1000 mL (or up to 50% of Total Water Intake) had benefits during memory tasks. Urine color and number of steps on the study day correlated with water consumed. The results suggest that a water-friendly environment supports school-aged children in adequate water intake resulting in better cognitive performance, especially short-term memory.

## 1. Introduction

Fluid consumption at school has received increasing attention regarding children’s health. Among other topics, drinking water and the effect of hydration on cognitive performance have gained interest [1]. Although a positive relationship between water consumption and cognitive performance is generally observed, the study results have shown many discrepancies. These depend in part on procedures used for the water intervention and on cognitive assessment tools.

Children are considered at greater risk of dehydration since their regulatory mechanisms are still insufficient compared to adults [2,3]. For example, physiological thermoregulation differs between children and adults. The underlying differences include lower sweat rates among children, but a higher ratio of body heat transfer per unit of body mass [4]. Children also have a higher total body water percentage than adults [5] and therefore need more fluids in relation to their body weight [5]. In addition, a balanced drinking behavior of children is important, particularly because children’s subjective feeling of thirst shows no correlation with fluid intake or hydration [6,7]. Measurements in healthy adult males showed that thirst only occurs when 2% of total body water is lost [8]. However, thirst sensation has not often been studied in children.

There are various laboratory tests to investigate hydration status. Urine markers, specifically osmolality, are helpful in determining changes in cell hydration and renal response to fluid intake. Some differences of an inter-individual renal response after water consumption have been reported in adults [9]. Only one study was found with healthy children aged 9–12 years that indicated a high variability in urine osmolality response after the same water supplementation [10]. The study also showed differences in cognitive performance associated with renal response after overnight fasting. This leads to uncertainty as to whether an adequate amount of fluids for children can be determined by physiological parameter such as osmolality. A helpful and simple method to assess the adequacy of children’s fluid intake may be urine color [1]. Adults, as well as children, can use this validated measurement method as a self-check to avoid elevated urine concentration [11].

The results of cross-sectional surveys from 13 countries (The Liq.In7 Initiative) revealed that 61% of children and 75% of adolescents did not consume enough water from fluids [12], regarding the recommendations for daily water intake (1.700 mL/day for boys 9 to 13 years old and 1.520 mL/day for girls 9 to 13 years old) provided by the EFSA [13]. These observations are consistent with further studies that showed that children drink insufficient amounts of water [14]. The individual recommended daily water intake based on the child’s weight was not considered for both analyses. The EFSA recommends higher total water intake (TWI) per day for boys than for girls 9 to 13 years old: 44 mL/kg versus 39 mL/kg, respectively [13]. In addition, children’s hydration status at school-start and over the school day seems to also be inadequate [15,16,17,18,19].

Despite the homeostatic mechanisms of the human water balance, even mild loss of total body water content can impair cognitive abilities [2]. Intervention studies showed that children’s cognition may benefit from drinking water. After drinking 300 mL of water versus no water, short-term memory was significantly better [20], similar to another study [19]. Other authors described a significant improvement after drinking supplementary water compared to no water condition in visual attention tasks [21]. New findings from a cross-over intervention study for 4 days appeared recently [22]. The results showed advantages in children between 9 and 11 years after a daily water consumption of 2.5 L of water on the color-shape switch task compared to the water condition ad libitum or low to 0.5 L. Therefore, the drinking behaviors of children during school days seems relevant for cognitive performance. According to the review from 2019, the first data collection on the hydration in healthy school children related to computerized cognitive tasks was carried out only a few years ago [23]. Studies with children in this context are still scarce and effects on a broader range of cognitive functions have not been studied so far. No study with children focused on urine color or thirst related to executive functions with computer tasks.

To fill the aforementioned scientific gaps, we focused on examining executive functions after water consumption in the morning including factors that affect drinking behavior, such as thirst and physical activity. To our knowledge, no studies have examined children’s executive performance in the school environment in this context using a computerized assessment tool and a subjective measurement of the hydration status. Thus, in the present cluster-randomized intervention trial CogniDROP (Cognition, Drinking Observation and Physical Activity), we aimed to investigate: 1) the short-term effects of increased water intake on various parameters of cognitive performance in children at school; 2) levels of hydration on the basis of two subjective criteria as reported by the children: urine color (Ucol) and thirst; and 3) include differences of physical activity during the test day.

## 2. Materials and Methods

### 2.1. Study Design and Recruitment

CogniDROP was designed as a cluster-randomized controlled intervention trial. Data were collected between October 2018 to January 2019 excluding school holidays. Recruitment of participants of the 5^th^ and 6^th^ grade (fourteen classes) was undertaken at a comprehensive school in Gelsenkirchen, Germany, where nutrition intervention studies had been carried out successfully earlier [24,25,26]. Physical education-free days were chosen for the intervention. Due to the cluster design (classes), the classes were randomly allocated to the intervention or control group using a computerized random number list. Both groups received water ad libitum during 1 school day. The intervention group received a training session on healthy drinking behavior and was reminded to drink water on the study day. The control group did not receive any training before the study and there was no active motivation to drink. The study staff informed all parents/guardians and respective teachers about the study procedures at a school assembly. Written informed consent of the parents/guardians and children was required for participation.

The study was approved by the Ethics Committee of the Faculty of Sport Science of Ruhr University Bochum (EKS V 22/2018) and was conducted in accordance with the Declaration of Helsinki.

### 2.2. Participants

All 400 children from the 14 5^th^ and 6^th^ grades classes were invited to participate on 1 intervention day, in a total of seven classes in each grade. Exclusion criterion was diagnosed learning disorder. Finally, 279 (68.8%) participants (mostly aged between 10–12 years) confirmed their participation by signing the consent form. All participants who completed the study day received a small toy. Participants were excluded from analysis if they did not complete the study procedures for reasons such as not consuming any water or absence (Figure 1).

### 2.3. Study Schedule and Treatment

The water supplementation intervention was integrated in the usual school procedures. The environmental temperature during the intervention days at noon differed between October and January (10.5 °C ± 5.3 in outdoor areas, 20.7 °C ± 0.7 in classrooms). The day before the water supplementation, the intervention group received a training session on healthy drinking behavior during the regular school curriculum. Body weight (lightly clothed, without shoes) was measured on an independent day close to the intervention day using a Seca 862 digital scale with graduation of 50 g below 150 kg (Seca Corporation, Hamburg, Germany). Physical activity expressed in step counts was measured with the ActiGraph GT3X accelerometer (Actigraph, Pensacola, FL, USA) worn on the wrist 1 day before and during the intervention day, in total 24 h.

The water supplementation day started at 8:00 when participants received a 500 mL scaled bottle (720°DGREE, Paderborn, Germany) with water. Water ad libitum was offered in front of the classroom with easy access for all children. The intervention classes were reminded to take their bottle and to drink water during common school breaks (four breaks between lessons). The study staff did not remind the children in the control classes. Study staff refilled the bottles of both intervention and control classes upon request, and only when the bottle was empty. The amount of water consumed was recorded by the study staff throughout the morning until the cognitive test. Remaining bottle water was weighed using a kitchen scale (MWF, Geislingen, Germany). Following the school schedule, regular food consumption was allowed. Before lunch break (at 12:00), participants self-reported urine color (Ucol) was evaluated based on a valid method to assess hydration in children of age 8–14 years in non-laboratory settings [11]. The participants within each class were instructed how to use the ‘8-point urine color chart’ (Human Hydration, LLC, Virginia, USA) and asked to urinate in dedicated washrooms (bright rooms with windows, white ceramics). Afterwards, the subjects were requested to assign the color that best describes their Ucol to the colors on the chart from 1 (lightest) to 8 (darkest). The study staff collected the charts. Afterwards, children’s cognitive performance was assessed by four computerized tasks performed in a quiet area at school within the class community.

### 2.4. Cognitive Assessment

Corresponding to our previous studies [24,26], a test battery was used, which was developed by the Institute for Work, Learning and Ageing (ALA) in Bochum, Germany, and adapted to the study conditions. Overall, four cognition tasks were performed in the sequence task switching, corsi block-tapping task, 2-back task, and flanker task (Figure 2). Speed and accuracy of the reactions were evaluated. The session began with a pretest session and a following low activity break (about 5 min). Then, the participants were asked to perform all tasks consecutively. The test battery was finished by the questions ‘*Are you thirsty?*’ and ‘*Did you have anything to drink at home this morning?’.* Incomplete cognitive measurements were recorded as missing data.

***Task Switching***. Spatial attention and switching abilities between two different tasks were measured as non-switching and switching responses. The stimuli on the screen were disordered numbers and letters in the boxes which were clicked with the mouse. Participants completed two non-switch trials (numbers and letters separately), followed by one switch trial (numbers and letters mixed). In trial 1, the children were asked to click the numbers from 1 to 26 in ascending order as quickly and accurately as possible. Trial 2, consisting of letters from A to Z, was carried out in alphabetical order. Trial 3 included both numbers (1–13) and letters (A-M) and participants were asked to click the targets in the correct order, but alternating between numbers and letters (e.g., 1-A-2-B…). With a correct response, boxes turned green or red for every wrong reaction. Each trial was limited to 3 min. Reaction time (RT) for non-switch and switch trial in seconds (s) was measured. Switching costs were defined as difference in response time between the switch trial and the non-switch trial [26] and was calculated as follows:
Switch Costs = Switch RT − Numb 26RT − (Lett 13RT − Numb 13RT)
where ‘Switch RT’ represents the mean RT for the visual search in the switch trial, ‘Numb 26RT‘ the mean RT for visual search numbers until 26, and ‘Lett 13RT‘ and ‘Numb 13RT’ represent the mean RT for the visual search of the first 13 targets in the non-switch trial.

Negative switch costs were excluded from the analysis (switch costs represent a positive respond time) [26,27].

***Corsi Block-Tapping Task.*** Performance on visual-spatial attention and working memory was tested in a forward tapping task. The reproduction of three to six block sequences was used to assess the children’s ability to remember the spatial order of the stimuli. Nine blue squares appeared on the screen and changed color in a random order. The task started with a three-box sequence, and after presentation, the participants had to click each of the boxes in the same order with a computer mouse. After each sequence, the children got feedback. A total of 12 block sequences were presented with increasing lengths: three, four, five, and sixboxes, three times each. The maximum length remembered was employed as the longest path. The number of correctly reproduced sequences was defined as correct immediate block span. In addition, a score was calculated for each sequence length. One to four points were awarded depending on the sequence length: one for the three-box sequence, to four for the six-box sequence. A higher score indicated better performance.

***2-Back Task.*** The ability to store and retrieve new information was examined. Short-term working memory performance and updating was assessed by the visual domain. One-hundred-and-six stimuli (images of fruit and vegetables) were presented consecutively on a computer monitor. Children were instructed to press a defined button each time a stimulus matched a stimulus two trials before. The 2-back-condition consisted of 21 correct trials. Each stimulus was presented for 500 ms with an interval of 2100 ms regardless of whether the participant responded within the limited time of 1400 ms or not. Feedback was given in the case of an error or correct response. RT was calculated only for correctly responded trials. The measure of accuracy was the ratio of false alarms (response to wrong trial) and the ratio of missing (no reaction to correct trial) as well as the sum of all correct events (total number of correct responses and the number of no responses to wrong trial).

***Flanker Task.*** The task measured inhibitory control whilst irrelevant stimuli were suppressed [24]. The task was characterized by presentation of congruent, incongruent and no-go trials. Directional or neutral target stimuli (triangle or circle) appearing at the center of the screen were displayed within confounding variables (flankers vertically-arranged). In the congruent condition, the central triangle was flanked by stimuli pointing in the same direction. Incongruent trials included flanking triangles that pointed in the opposite direction compared to the central triangle. In the no-go trials, a circle appeared in the middle. The participants were asked to respond to the central triangle by pressing a defined computer key with the right or left index finger according to the direction of the target or to not respond to the circle.

Within 102 trials, 35 congruent, 35 incongruent, and 32 no-go target stimuli were presented. Stimulus intervals of 1000 ms (± 20%) started with the top and bottom flankers for 100 ms followed by another 100 ms together with the target in the middle. The maximum RT was 1200 ms, but a feedback *faster* occurred in the event of a response later than 600 ms. The count of false alarms in no-go trials was recorded as well as the RT slowing (difference between mean RT of correct reactions in incongruent and congruent trials) and difference error rate (difference between ratio of incorrect reactions in incongruent and congruent trials). To avoid unplausible results (e.g., due to playing with computer buttons and ignoring the instruction), a negative RT slowing and a negative difference error rate were excluded [24].

### 2.5. Statistical Analyses

A sample size calculation was performed based on a power of 80%, α of 5%, and Cohen’s *d* effect size of 0.50 (G*Power for Windows, version 3.1, Duesseldorf, Germany). A sample size of 51 participants was needed per group to detect significant differences within groups for outcomes.

All analyses were performed using the statistical software package IBM*SPSS* Statistics for Windows, version 25.0 (IBM Corp., Armonk, NY, USA). The level of significance for all analyses was set at *p* ≤ 0.05. Characteristics of participants were reported as frequencies and percentages for categorical data. Cognitive outcomes were primary outcomes of the study. Median, 25^th^ and 75^th^ percentile were calculated for all cognitive variables, and distributions were tested for normality with the Shapiro–Wilk test. Differences between the intervention and control groups were tested with an independent samples *t*-test for normally distributed data or the Mann Whitney-U-Test for non-normally distributed data. Before further analysis, locally weighted scatterplot smoothing (LOESS) techniques were used to visualize the strength and direction of relationship between water consumption and cognitive outcome.

In the total sample, independent of group allocation, the amount of water consumed was categorized. Perry and colleagues [10] tested cognitive performance under experimental conditions, providing 750 mL of water over a period of 2 h. Therefore, we expected a maximum water consumption of 1500 mL over a period of 4 h in this study. Based on studies in children showing relations between urinary osmolality and total water intake (including food) for breakfast above or below 500 mL [18], the following categories were classified: group 1: < 500 mL; group 2: 500 ≤ and < 1000 mL; group 3: 1000 ≤ and < 1500 mL; group 4: ≥ 1500 mL. The Kruskal–Wallis test was used to determine statistically significant differences in cognitive performance between the categories. For analyzing the multiple pairwise comparisons, Dunn’s test with Bonferroni correction was performed. For evaluating the strength of a significant difference between categories, the effect size was calculated ($r=Z/\sqrt{n}$) [28]. Spearman coefficient was used to correlate Ucol and cognitive performance. For the cognition comparisons between children who were thirsty and those who were not, Mann–Whitney U Test test for non-normally distributed data was used. Further parameters, such as children’s body weight and the number of steps during, as well as before the test day (from 08:00 to 13:00 hrs on test day, the last 24 h till 13:00 on test day), were included in the analysis. The individual recommended daily water intake was calculated based on the child’s weight and compared with documented measurements of water consumption (estimated water intake without including food). The test for partial correlation was used to measure the relationship of an intervention effect whilst controlling for the secondary outcomes such as physical activity expressed in step counts. Intention-to-treat analysis (ITT) based on available cases was used for all existing data.

## 3. Results

### 3.1. Participants

Figure 1 shows the study flow chart. Four children with learning disabilities were excluded from analyses. Two-hundred-and-fifty children participated in the study and followed the study instructions.

The participants’characteristics are presented in Table 1. Of the 250 children, 96 (38.4%) were female, 117 (46.8%) in the 5^th^ grade, and the mean age was 10.9 ± 0.8 years. There were no differences in grade, sex, age, and water intake during the intervention morning at school between both intervention and control group. The average water intake of all children was 1175 ± 640 mL. In total, boys drank 1228 ± 686 mL of water vs. girls 1091 ± 552 mL, but the difference was not significant (*p* = 0.235). This corresponds to about 70% of the daily recommendations by EFSA for water intake for girls and boys. Moreover, about a third of the children were thirsty. Subjective thirst at midday was reported significantly more frequently in the intervention group (*p* = 0.033). Children in the intervention group indicated more often a significantly lighter Ucol on the eight-point urine color chart (*p* = 0.037), with a frequency of 76.7% in scale ‘1‘ vs. 66.7% in the control group. Two-hundred-and-forty-eight children answered the question: *Did you have anything to drink at home this morning*. Overall, 25 (18.7%) children in the intervention group and 14 (12.3%) in the control group started the school day without drinking (*p* = 0.220). Morning water consumption at school was not different between children who skipped drinking before school or not (1349 ± 770 mL vs. 1151 ± 612 mL, *p* = 0.191). Most children (*n* = 245) had data on physicial activity. They took an average of 4755 ± 757 steps during the morning, while total step counts over the 24-h period were 15617 ± 2854. Differences of yielded steps between both study groups were not significant (Table 1).

### 3.2. Cognition

Results for 2-back task were missing for two participants out of the 250 included. In total, 25 had to be excluded for task switching analysis because of negative switch costs and 62 for flanker task because of negative RT slowing and difference error rate.

The performance of children in the intervention group did not differ significantly from children in the control group in any of the cognition measurements (Appendix A). The results did not change for flanker task when all children were included (Appendix A).

### 3.3. Differences Between Water Categories

As total morning water consumption did not differ between the intervention and control group, we examined cognition differences between the water consumption categories in the total sample (*n* = 250). Results in Table 2 demonstrate that the cognitive performance differed between the four water consumption categories. There were significant differences between the four independent groups in switch task and in corsi block-tapping task.

In general, children achieved better results if they drank more than 0.5 L of water during the morning. No significant cognitive advantages were found for water consumption over 1.5 L for these parameters.

Significant differences were found between water categories for all outcomes in corsi block-tapping task: longest path (*p* = 0.049) immediate block span (*p* = 0.004) and score (*p* = 0.004). After applying the Bonferroni correction, a significant difference persisted only in immediate block span (*p* = 0.026) and in score (*p* = 0.025) between the second and fourth water category. The mean effect size of these differences was small to medium for both immediate block span and score (*r* = 0.23).

In the switch task, only results of visual search switch were significantly different among the water categories (*p* = 0.037) (Table 2). However, the pairwise comparisons in a Dunn–Bonferroni post hoc manner did not indicate any significance.

### 3.4. % of Estimated Individual Recommended Water Intake and Step Counts

Using recommended values of daily water intake (44 mL/kg for boys and 39 mL/kg for girls) and recorded children’s body weight (range between 26.6 and 99.0 kg), the individual estimated recommendation for daily Total Water Intake (TWI) was calculated for 221 children (range between 1049 mL and 4356 mL). There was no relationship between recommended daily TWI and morning water volumes (*p* = 0.069). For the total sample, the actual water intake estimated in the morning corresponded to the average of 65% ± 37.8 of the daily TWI. Overall, 131 (59.3%) of 221 children reached the recommended 50% of the TWI in the morning. According to the children’s individual weight, measured water intake as a percentage of TWI was used for further analysis and categorized by intervall of 25% (group 1: ≤ 25%; group 2: 25% < and ≤ 50%; group 3: 50% < and ≤ 75%; group 4: > 75%). Table 3 shows results for the corsi block-tapping task and four categories of percent of recommended water intake. Similar to the categories for the morning volume of water, significant differences were found between four water categories in a percentage of TWI for all outcomes in corsi block-tapping task (Table 3). No significant differences were found for other cognitive parameters in this context (*p* > 0.05).

Comparisons between subgroups regardless of the Dunns test with Bonferroni correction verified significant differences between categories up to 50% and over 75% of TWI for immediate block span (*p* = 0.004, *r* = 0.26) and score (*p* = 0.004, *r* = 0.26).

As shown in Figure 3B, some children achieved more than 100% of the daily TWI in the morning (without considering water from food). The individual estimated water intake in percent correlated with step counts for the 24-h period, but not for the activity in the morning (Figure 3). On the other hand, there was a small correlation between physical activity expressed in steps during the school day and water amount in mL (*p* = 0.006, *r* = 0.18), as well as between step counts for the 24-h period and water amount in mL (*p* = 0.004, *r* = 0.19).

### 3.5. Correlation Analysis

Based on a Loess curve, cut-off for 1000 mL of water intake was defined for further analysis. Spearman correlation analysis revealed a relationship between water intake up to 1000 mL and the longest path in the corsi block (*p* = 0.026, *r* = 0.20) (Figure 4), as well as the ratio of missings in the 2-back task (*p* = 0.007, r = −0.24) (Figure 5). There was a negative correlation for parameters in 2-back task and water range above 1000 mL (correct events *p* = 0.044, *r* = −0.17). In a similar way, the cut-off threshold for the percentage of the individual TWI was defined. For the value below 50%, no significant correlation was observed for the parameters in the corsi block-tapping task (Figure 4), but a significant correlation was observed between the estimated water intake below 50% and ratio of missings in the 2-back task (*p* = 0.018, *r* = −0.25) (Figure 5). No correlation was found for the value above 50% water and the cognitive outcomes. Based on the test for partial correlation, step counts for the 24-h period did not influence any results of bivariate correlation between water intake and cognitive performance (*p* > 0.05). The significance of the relationship between water consumption below 1000 mL and cognition decreased while taking away the effects of step counts on longest path correction in the corsi block (*p* = 0.035, *r* = 0.19) and ratio of missings in the 2-back task (*p* = 0.015, *r* = −0.22). The relationship between correct events in 2-back task and amount of water above 1000 mL disappeared when the effects of step counts were removed. According to cut-off threshold for the percentage water intake, partial correlation after controlling for steps showed a small improvement in the relationship between water intake below 50% of daily TWI and ratio of missings in 2-back task (*p* = 0.013, *r* = −0.27).

### 3.6. Thirst and Step Counts

Between thirsty children and children who reported feeling “not thirsty”, the mean water consumption during the school day was not different (1192 mL vs. 1175 mL, *p* = 0.873). An average of 62.0% of daily TWI was estimated for children with thirst and 66.8% without thirst (*p* = 0.796). Accelerometer-based amount of steps on study day did not differ (*p* = 0.205) between children who were thirsty (4683 ± 712 steps) or not (4802 ± 776 steps), as well as the 24-h period of step counts showed no differences (thirsty children 15762 ± 2692 steps, not thirsty children 15528 ± 2959 steps, *p* = 0.508). There was no significant sex difference in self-reported thirst (41.7%, *n* = 40 girls and 31.6%, *n* = 48 boys reported thirst, *p* = 0.134). Ucol did not differ between thirsty and non-thirsty children (*p* = 0.491). Children who did not feel thirsty were faster in the switch task (visual search letters, 37.5 ± 12.5 s vs. 41.7 ± 16.8 s, *p* = 0.029 and visual search switch; 90.3 ± 26.1 s vs. 99.2 ± 28.2 s, *p* = 0.017).

### 3.7. Urine Color

According to the bivariate correlation for non-parametric tests, there was a significant correlation between the amount of water consumed and Ucol (*p* < 0.001, *r* = -0.27) for all children, but there was no significant relationship between step counts on test day and Ucol (*p* = 0.517). Ucol did not differ between sex (*p* = 0.869). There were no significant relationships between Ucol and any of the task performances (Appendix A).

## 4. Discussion

In this study, we investigated the short-term impact of water consumption during a school day on cognitive performance in school-aged children with intervention by active motivation to drink water or not. We found neither differences between the intervention and control group in the amount of drinking water, nor differences in cognitive outcomes. Thus, short-term education on healthy drinking behavior in this study did not result in different drinking behaviors between both groups. Studies showed that school nutrition education alone does not affect children’s eating habits and that adult strategies to influence older children‘s food preference are limited [29]. However, on average, both the intervention and control group in this sample reached the recommended amount of water for the morning [13]. The results therefore indicate a better water supply for children in our sample compared to observational studies worldwide [12]. Merely, the fact that water was easily accessible caused an increased water intake in both groups, independently of active motivation. This is in line with other studies [29,30] showing that the free access to water has the most important impact on children’s drinking behavior. In the prospective „trinkfit“ interventional study in the elementary school setting in Germany, Muckelbauer and colleagues [31] also supported the assumption that the availability and accessibility of water in school settings affects drinking behavior. Children’s increased water consumption was observed in this study after provision of water fountains and water bottles for a year. Considering the results, this study supports an effective strategy for increasing the intake of water through a free and easily accessible water source.

Furthermore, we observed a non-linear relationship between the water intake and cognitive performance. This resulted in the positive relationship between water consumption up to 1000 mL and better results in memory tasks. A subsequent decreased performance in 2-back task for water consumption over 1000 mL was observed. Consuming 50% of the recommended daily TWI throughout the morning led to increased performance in 2-back task. We no longer saw significant cognitive benefits after consumption of more than 50% of the daily TWI in the morning. Based on an analysis of pairwise comparisons, we suggest that spatial memory performance was considerably worse if the children drank more than 1.5 L of water as well as above 75% of daily TWI compared to the water category of less than 1 L or less than 50% of the daily recommendations. As a result of the increased morning amount of water compared to the recommended daily amount for this age group, a reversal of the effect is possible [32,33]. There is no exact explanation of how children respond to changes in water intake. Studies in adults have examined many possible mechanisms, which involved control of brain volume during hypoosmolar conditions and the levels of intracranial blood. It has been shown that not only dehydration, but also at some stage, the increased water intake could result in adverse effects [2,34,35]. It must be emphasized that the small number of studies with children makes it difficult to interpret the results.

In addition, these analyses suggest that children in the water category lower than 0.5 L or below 25% of daily TWI performed the cognitive tasks worse than children in the other categories. Unfortunately, it was not possible to substantiate this statement with the post hoc analysis. The small number of subjects in the lowest water category (*n* = 23) should be taken into account. Fifty-one children were needed to determine the expected cognitive differences between all water categories. However, the correlation analysis showed that increasing the water intake to 1000 mL or 50% of daily TWI, respectively, had a positive relationship on cognitive parameters in the memory tasks. Less conclusive results were observed in the switching task, in contrast to other reports [22]. A high degree of variability in the measurement tools and treatments may lead to contradictory results. No association was found in flanker task, which seems to be less sensitive to changes in water balance [22]. Given these varying influences on the different cognitive parameters and water consumption, different neural mechanisms could be responsible for the observed performance. The first study testing a hierarchical model of inhibitory control suggested dependent neurobiological mechanisms in the prefrontal cortex consisting of neural “top-down representations” [36]. The authors emphasized the dependence of response inhibition and visual selective attention on working memory capacity as a “higher-order representation”. It could be suggested that a higher-order cognitive ability, such as working memory, is more sensitive to changes in the water balance. This knowledge may explain the stronger response of working memory to water consumption in this study. Thus, we support the results of other studies in children [19,20] that an adequate increase in water intake may improve performance in memory tasks. When comparing our results with these studies, we tested short-term memory after a 4-h water intervention period, while others observed a positive effect on short-term memory after acute drinking 300 mL of water or water intervention from morning to afternoon.

Based on findings in adults and children describing that the regulatory mechanisms of the body’s fluid balance may vary between individuals [32,34,37], we conclude that over the morning the same amount of water consumption influenced cognitive performance differently. Differences in the renal regulation and response to water intake may originate from different fluid requirements due to body weight as well as age [5,32]. In addition, physical activity and the associated water loss could play an important role [1,32]. According to Perales-García et al. [38], schoolchildren had an increased risk of dehydration if they were more physically active because they did not replace fluid loss. Our results showed a correlation between physical activity and water consumption, not only during the morning at school but in the past 24 h. We conclude that fluid loss from physical activity was balanced out by increased drinking at school without significant losses in cognitive performance. However, increased step counts could result from repetitive trips to the restroom because of increased drinking. On the other hand, the restrooms in the school are near the classrooms and only a short distance is required to reach them. Therefore, it seems unlikely that an increased number of trips to the restroom led to an overall higher number of step counts.

According to thirst sensation, our results showed that having subjective thirst was not explained by water intake, urine color or physical activity. This observation might suggest less effective thirst mechanisms in children according to another report [32] or a positive water balance in our study population, potentially due to the reached recommended amount of water intake. It is possible that children notice thirst only with increased total body water deficits [6]. Moreover, it was unclear why children in the intervention group reported thirst more often compared to the control group. The two groups did not differ in the amount of drinking water. There is a socio-psychological explanation for behavior to meet researchers’ expectations, the Hawthorne effect [39]. The study staff had more encounters with children in the intervention group, which could lead to a different relationship and trust. This could affect the manipulations in self-assessment of thirst. However, we found no associations between memory performance and thirst, in line with previous studies in adults [40,41]. Children who were not thirsty reacted faster in the task switching. Their response in visual search letters and visual search switch was shorter than for thirsty children. Previously, authors observed an effect of water consumption on a visual search task and thirst, but independently of each other [40]. They suggested that thirst in children did not moderate the relationship between drinking water and cognitive performance. Others [7,42,43] reported that self-reported thirst in children was not correlated with urine specific gravity and urine color. Thus, it is not clear how sensitive cognitive performance is to feeling thirsty in children. Studies in adults reported individual differences in the perception of subjective feelings of thirst regardless of water consumption [7,42,43]. Research with healthy school-age children is limited and therefore needed [11].

This study also showed that higher water consumption was associated with a brighter color on the urine scale. In contrast, we observed that the intervention group reported brighter urine color although both the intervention and control group consumed the same amount of water on average. This may be explained by the fact that the intervention included a training session on healthy drinking behavior including information on impact on Ucol. This could, in individual cases, lead to an under- or overestimate of the self-report. Because of the overall high water intake within 4 h and probably no signs of dehydration in the study population, no relationship between Ucol and cognitive outcomes was not surprising.

Finally, studies investigating environmental factors such as environmental temperature showed that beverage consumption does not differ in quantity during winter and summer. However, the seasons influence the quality of the drinks and the choice of food [44]. On the other hand, hydration status differs according to composition of foods and beverages [18,45,46]. Therefore, seasonal differences in food intake in this study population may be possible between October and January, and could have resulted in differences in the response to water intake.

## 5. Strengths and Limitations

There were several strengths of the present study. First, this study is novel because of the implementation of well-established computerized tasks to test the executive function after voluntarily drinking water at school. Second, we measured the volume of water consumed throughout the morning and we examined differences in water intake with regard to physical activity and body weight. Additionally, all procedures were integrated into regular school life as closely as possible. Thus, we were able to test children’s cognitive abilities in the natural environment.

However, we acknowledge some study limitations. This study was carried out in a school setting and therefore with limited diagnosis of dehydration, which would be desirable to strengthen the analysis. Further, the low number of subjects in the lowest water category may lead to less sufficient power to detect significant differences between the water categories in terms of cognitive performance. Moreover, we tried to standardize the measurement of water and handed out identical water bottles to all children at the school beginning. This procedure might have further motivated all children to drink water. It is therefore unknown if the the control group drank more water than usual. We also cannot rule out that the two groups influenced each other, since it was not possible to separate groups at school. We also did not record food consumption during the test day and could not consider this when estimating the TWI. Additionally, it should be mentioned that self-reported urine color and thirst could contain some distortions.

## 6. Conclusions

In conclusion, the current intervention study indicated for the first time short-term effects of water provision at school on different executive functions based on computerized cognitive tasks. The mean effect size of these effects was small to medium. Increasing water intake improved children’s memory in particular. Furthermore, most children in the study population achieved the recommended water intake. Weak or unrelated relationships were found between water intake, urine color, thirst, and step counts. The subjective measurement of hydration including urine color and thirst is a practical tool for children but less helpful with an already sufficient water intake. Replication of the current findings using more valid assessment of dehydration is required.

## Figures and Tables

**Figure 1 nutrients-12-01297-f001:**
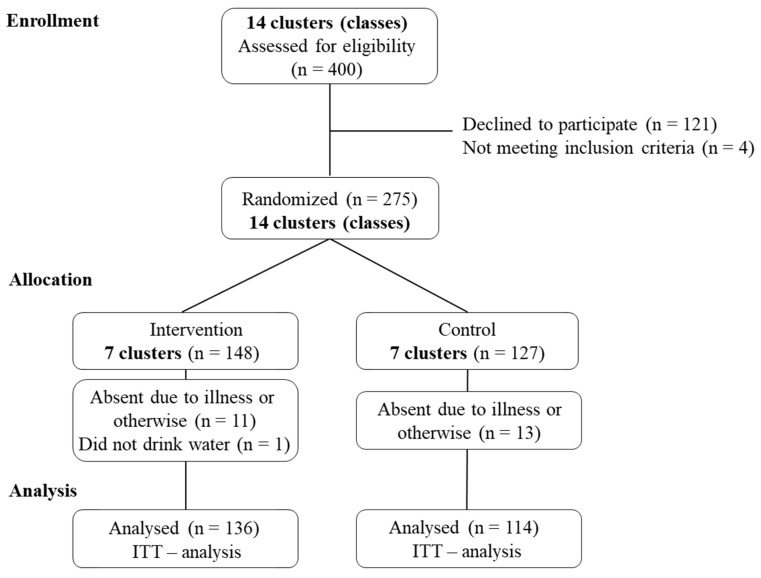
Flow chart for population recruitment.

**Figure 2 nutrients-12-01297-f002:**
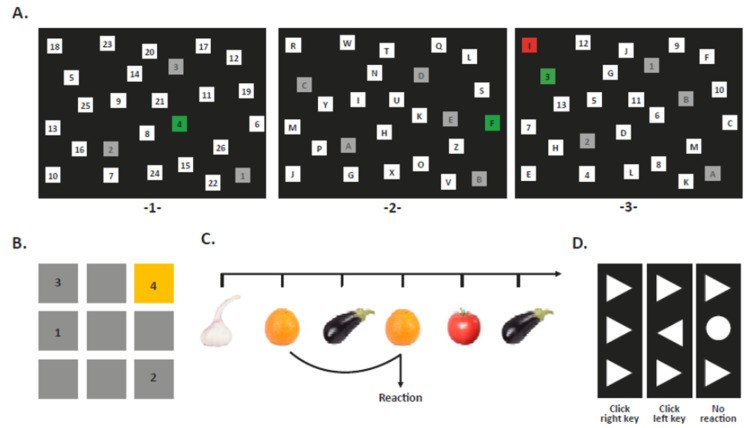
Computerized cognitive task. (**A**) Visual attention and task switching measured by switch task. The task comprised of three sections. 1. First section, numbers (non-switch). Numbers had to be click in ascending order with the mouse curser. 2. Second section, letters (non-switch). Letters from A to Z had to be clicked alphabetically. 3. Third section, number and letters (switch). Numbers and letters had to be clicked alternately in ascending order (i.e., 1-A-2-B-3-C…). (**B**) Visual-spatial attention and working memory measured by corsi-block tapping task. Forward sequence of blocks was displayed and gradually increased in length up to six blocks. The sequence in the order had to be repeated. (**C**) Working memory updating measured by 2-back task. Fruits and vegetables were displayed on a computer screen. A predefined key had to be pressed when the current image was the same as the image two trials back. (**D**) Inhibitory control measured by flanker task. Directional response to the targets: congruent flankers and incongruent flankers were needed by pressing a defined computer key.

**Figure 3 nutrients-12-01297-f003:**
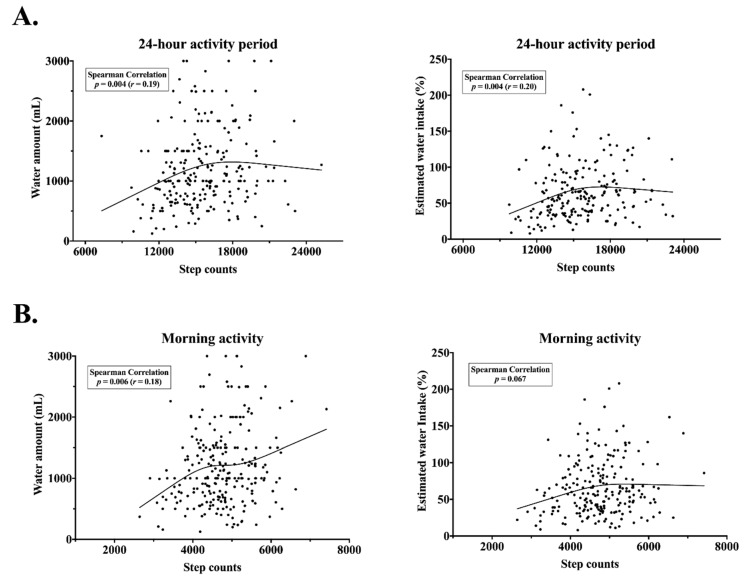
Scatterplots of a relationship between physical activity and water intake during a school day. (**A**) Non-linear positive association for step counts over the 24-h period and water amount (*n* = 228)/percentage of individual estimated water intake in the morning (*n* = 204) (**B**). Non-linear positive association for step counts throught the morning and water amount (*n* = 244). No association for step counts throught the morning and percentage of individual estimated water intake (*n* = 218).

**Figure 4 nutrients-12-01297-f004:**
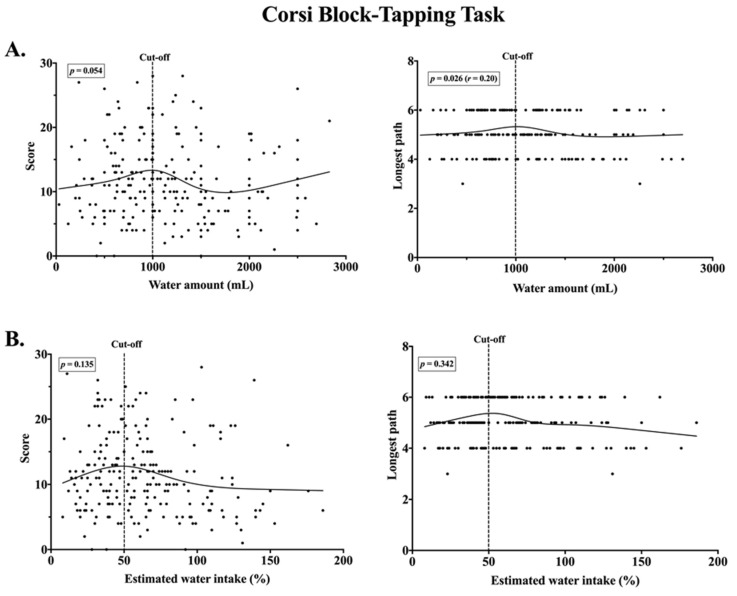
Scatterplots of a relationship between water intake during a school day and cognitive performance in the corsi block-tapping task. (**A**) No association for score and water amount below cut-off (*n* = 131). Non-linear positive association for longest path and water amount below cut-off (*n* = 129). (**B**). No association for score/longest path and percentage of estimated water intake below cut-off (*n* = 92/*n* = 90).

**Figure 5 nutrients-12-01297-f005:**
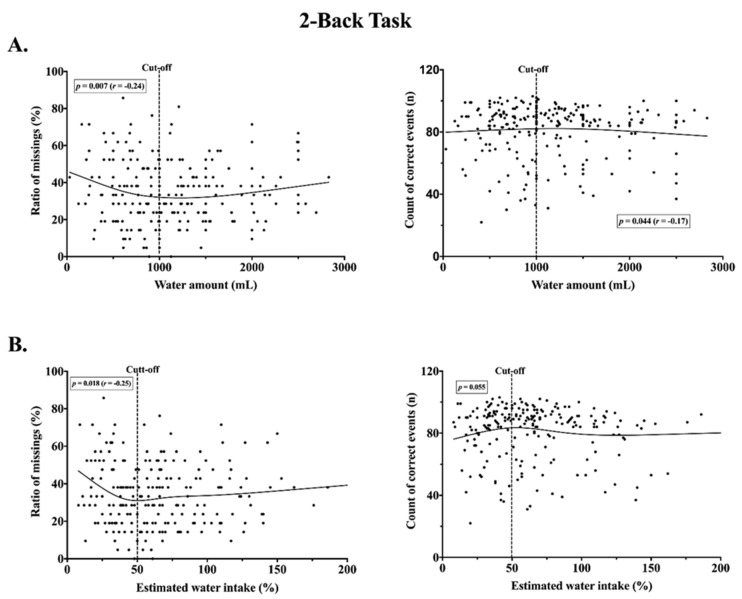
Scatterplots of a relationship between water intake during a school day and cognitive performance in the 2-back task. (**A**) Non-linear negative association for ratio of missings and water amount below cut-off (*n* = 107). Non-linear positive association for correct events and water amount over cut-off (*n* = 107) (**B**). Non-linear negative association for ratio of missings and percentage of estimated water intake below cut-off (*n* =90). No association for correct events and percentage of estimated water intake below cut-off (*n* = 90).

**Table 1 nutrients-12-01297-t001:** Sample characteristics (*n* = 250).

	Intervention	Control	*p* Value
Total n (%) ^1^	136 (54.4)	114 (45.6)	0.164
Boys	86	68	0.147
Girls	50	46	0.612
Grade 5	63	54	0.405
Grade 6	73	60	0.227
Age ^2^ (years), mean ± SD	11.0 (0.8)	10.8 (0.7)	0.075
Water consumption ^2^ (mL), mean ± SD	1204 (639)	1141 (642)	0.398
Rating of Ucol-1 ^2^, n (%)	104 (77.6)	76 (66.7)	0.037
Self-reported thirst ^3^, n (%)	55 (40.4)	32 (28.1)	0.033
Step counts 8-13 ^4^, mean ± SD ^a^	4760 (761)	4745 (758)	0.883
Step counts 24-Recall ^4^, mean ± SD ^b^	15430 (2831)	15855 (2891)	0.266

^1^ Chi-Square test, ^2^ Mann–Whitney U test, ^3^ Fisher’s Exact Test, ^4^
*t*-test, Ucol: urine color, Ucol-1: the lightest color on the chart, *n*: number of participants, ^a^
*n* = 221; ^b^
*n* = 228, *p* ≤ 0.05.

**Table 2 nutrients-12-01297-t002:** Results of the cognitive tasks by water consumption categories in the total sample (*n* = 250).

Tasks	< 0.5 L			0.5 ≤ and < 1.0 L			1.0 ≤ and < 1.5 L			≥ 1.5 L			
	$\tilde{x}$	25^th^	75^th^	$\tilde{x}$	25^th^	75^th^	$\tilde{x}$	25^th^	75^th^	$\tilde{x}$	25^th^	75^th^	*p* Value
**Switch Task**		***n* = 21**			***n* = 73**			***n* = 66**			***n* = 64**		
Switch costs (s)	33.6	22.0	58.6	27.5	19.4	41.2	**24.6**	13.3	38.7	31.1	19.2	52.2	0.084
Visual search letters (s)	40.0	31.1	48.1	35.0	30.4	44.2	**34.4**	30.3	39.7	36.1	31.7	43.0	0.298
Visual search numbers (s)	56.7	46.6	60.9	**51.0**	45.8	59.8	52.6	45.0	61.9	53.8	46.1	64.3	0.723
Visual search switch (s)	98.4	81.7	121.2	**83.3**	73.1	103.6	86.0	70.1	103.2	97.1	78.4	120.6	0.037
**Corsi Block**		***n* = 23**			***n* = 82**			***n* = 73**			***n* = 71**		
Longest Path (n)	5.0	4.0	5.3	5.0	5.0	6.0	5.0	5.0	6.0	5.0	4.0	6.0	0.049
Correct immediate block span (n)	5.0	4.0	6.3	**7.0**	5.0	8.0	6.0	4.5	8.0	5.0	4.0	8.0	0.004
Score	8.5	6.0	12.0	**12.0**	9.0	18.0	**12.0**	8.0	17.0	9.0	6.0	15.0	0.004
**2-back**		***n* = 23**			***n* = 80**			***n* = 73**			***n* = 71**		
Ratio of missings (%)	38.1	28.6	52.4	33.3	19.0	47.6	**28.6**	19.0	40.5	33.3	19.0	42.9	0.133
Ratio of false alarms (%)	14.1	7.1	28.2	11.8	7.1	25.9	**10.6**	6.5	19.4	12.9	9.4	25.9	0.228
RT (ms)	454	40.9	57.5	477	399	550	454	404	550	474	334	546	0.744
Count of correct events (n)	84.0	69.0	91.0	86.5	69.8	92.0	**90.0**	83.0	95.0	86.0	76.0	91.0	0.086
**Flanker Task**		***n* = 16**			***n* = 62**			***n* = 55**			***n* = 55**		
RT slowing (ms)	74.8	49.5	119.0	**74.6**	53.9	98.9	79.5	56.8	99.1	70.9	56.2	94.6	0.748
Difference error rate (%)	18.6	14.3	30.7	**17.1**	10.7	28.6	22.9	8.6	31.4	**17.1**	8.6	31.4	0.720
Count of false alarms (n)	12.5	3.5	23.3	7.5	3.0	16.0	**6.0**	3.0	13.0	9.0	4.0	17.0	0.095

Kruskal–Wallis test; data are presented as $\tilde{x}$ = Median, 25th and 75th percentiles. Results marked in bold represent the best average values; *p* ≤ 0.05.

**Table 3 nutrients-12-01297-t003:** Cognitive performance in corsi block-tapping task between the four water categories (% of TWI).

	≤25%				≤50%			≤75%			>75%		*p* Value
	$\tilde{x}$	25^th^	75^th^	$\tilde{x}$	25^th^	75^th^	$\tilde{x}$	25^th^	75^th^	$\tilde{x}$	25^th^	75^th^	
**Corsi Block**		***n* = 23**			***n* = 82**			***n* = 73**			***n* = 71**		
Longest Path (*n*)	5.0	4.0	5.0	5.0	5.0	6.0	5.0	5.0	6.0	5.0	4.0	6.0	0.009
Correct immediate block span (*n*)	5.0	4.0	6.0	**7.0**	5.0	8.0	6.0	5.0	8.0	5.0	4.0	7.0	0.009
Score	9.0	6.0	12.0	**13.0**	9.0	18.0	**12.0**	8.0	15.5	9.0	6.0	12.0	0.005

Kruskal–Wallis test; data are presented as $\tilde{x}$ = Median, 25^th^ and 75^th^ percentiles. Results marked in bold represent the best average values; *p* ≤ 0.05.

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
