# Peer review of "Water Consumption during a School Day and Children’s Short-Term Cognitive Performance: The CogniDROP Randomized Intervention Trial"

_nutrients, 2020, doi:10.3390/nu12051297_

Round 1
Reviewer 1 Report
Dear editor,
Overall, I liked the study. In summary, the authors examined whether children’s cognitive performance was affected by water consummation during a school day. The researchers compared two groups: a control and an intervention group, which was given education on healthy drinking behaviour. Unfortunately, the two groups did not differ in terms of water drinking and cognitive performance. As mentioned by the authors in the general discussion, a sensibilization to drinking behaviours at school might have occurred in both groups (including the control group).
Nevertheless, by grouping all students and splitting them according to the level of water consummation, the authors found that the level of water drinking influenced children’s cognitive performance. The manuscript is well-written and easy to understand. The statistical methods are correct (but see my detailed comments). I also liked the general conclusion.
Detailed commentaries
Page 2. 3rd paragraph.
The link between this paragraph and the previous paragraph is not clear.
page 2, 6th paragraph
The objectives are clear. However, what is not clear is how they are related. The introduction attempts to draw some links between these concepts but how important they are is missing. I am also concerned about the justification for using urine colour as a self-check measurement method in children.
2.3 Study schedule and treatment
I am also concerned about the possibility of transmission from one class to another during the study. The authors should discuss this possibility in the general conclusion. This possibility might explain why the two groups did not differ in terms of levels of water drinking.
2.5 Statistical analysis
Why did the authors use a one-tailed test to determine the number of participants? I think the authors should use a two-tailed test (0.05/2) when determining the sample size for a given level of power (0.80).
I wonder why the authors did not use a hierarchical model to analyze the data and compare the conditions. Given that children were nested within a class, it would be easy to use a multilevel model to analyze the data. This way, the analysis would be more sensible to the possible variations within and between classes.
Some correlations are very small (r = 0.18, r = 0.19). R-squared are very, very small. I am not sure that these correlations are relevant.
Depending on the outcome variable, a linear or non-linear mixed regression model, including water drinking levels, sex, urine colour, age and/or group, could be an alternative to the approach used and presented by the authors. However, I do not have access to the data, and maybe this is inappropriate.
I liked the general discussion. In short, this is a lovely study with some limitations.
Author Response
Response to Reviewer 1 Comments
We thank the reviewer for her/his thorough and careful evaluation of our paper and for the constructive critique, comments and suggestions. We considered all issues addressed and revised our paper accordingly.
1. Page 2, 3rd paragraph.
The link between this paragraph and the previous paragraph is not clear.
We agree with you. In the 2nd paragraph, we discussed the risk of dehydration in children depending on physiological thermoregulation and thirst. The following paragraph pointed out an individual urine osmolality response without previous explanation. We have therefore added a brief explanation of hydration biomarkers in the paragraph to improve the flow of the text. (page 2, line 17-20).
2. Page 2, 6th paragraph
The objectives are clear. However, what is not clear is how they are related. The introduction attempts to draw some links between these concepts but how important they are is missing. I am also concerned about the justification for using urine colour as a self-check measurement method in children.
Thank you for this suggestion. We initially aimed to test the effect of our intervention on water intake and hydration status on cognition after controlling for potential confounders, such as physical activity. Hereby, the limited number of studies that have examined children's executive function after acute water supplementation in the school environment should be extended. With regard to your comment, we have specified the objectives (page 3, line 1-5).
Laboratory-based assessment tools (i.e. urinary markers) are more expedient hydration assessment techniques to test the adequacy of water intake and hydration status. Unfortunately, due to school reluctance measuring urine osmolality was not possible. Therefore, we used a simple method to assess the adequacy of children's fluid intake by urine colour. This validated measurement in healthy 8-14 years old children has a strong diagnostic capacity to differentiate between euhydration and hypohydration (Kavouras et al. 2007). It is possible that the urine color self- assessment in children can be under- or overestimated. We mentioned this limitation in the discussion (page 16, line 35-42).
3. 2.3 Study schedule and treatment
I am also concerned about the possibility of transmission from one class to another during the study. The authors should discuss this possibility in the general conclusion. This possibility might explain why the two groups did not differ in terms of levels of water drinking.
You are right, there might be a possibility of transmission. By performing a randomized study within a school setting, we could not separate the intervention and control groups properly from each other, which might have contributed to a mutual motivational influence between the classes. In the discussion, we stated that children’s drinking behavior is complex and multifaceted. It cannot be assured that the motivation of voluntary drinking differed between classes (page 15, line 1-9). We added the restrictions for better understanding in the limitations (page 17, line 15-16).
4. 2.5 Statistical analysis
Why did the authors use a one-tailed test to determine the number of participants? I think the authors should use a two-tailed test (0.05/2) when determining the sample size for a given level of power (0.80).
For the sample size calculation we used a standardized G * Power calculation program based on estimated level of power 0.80, alpha of 5%, and Cohen’s d effect size of 0.50 instead of a hypothetical mean and standard deviation (Faul et al. 2007). This was because of the novel research and we had no comparable hypothetical measurements.
I wonder why the authors did not use a hierarchical model to analyze the data and compare the conditions. Given that children were nested within a class, it would be easy to use a multilevel model to analyze the data. This way, the analysis would be more sensible to the possible variations within and between classes.
Our choice of statistical methods based mostly on tests for non-normally distributed data. The requirements for the multi-level analysis is normal distribution for both levels of hierarchical models. These necessary conditions were not fulfilled (for example gender). In fact, the relationship between water consumption and cognitive parameters showed no linearity, nor monotony, so that regression/ hierarchical models were not possible (Th. Gries, 2016). Therefore, we suggest keeping this approach.
Some correlations are very small (r = 0.18, r = 0.19). R-squared are very, very small. I am not sure that these correlations are relevant.
We agree with you that some correlations are very small. Although we already mentioned it in the results section and in the discussion, we also added it to the conclusions (page 17).
Depending on the outcome variable, a linear or non-linear mixed regression model, including water drinking levels, sex, urine colour, age and/or group, could be an alternative to the approach used and presented by the authors. However, I do not have access to the data, and maybe this is inappropriate.
While, given the complexity of these modelling approaches, the coverage can not, of course, be exhaustive. In the present analysis, we primarily focused on the relationship between the amount of water consumed and cognition considering non-linear and non-monotonous dependency of the variables. Further analyzes as part of the study are in preparation. We will take your comments into account.
References
Faul, F., Erdfelder, E., Lang, A.-G., & Buchner, A. (2007). G*Power 3: A flexible statistical power analysis program for the social, behavioral, and biomedical sciences. Behavior Research Methods, 39, 175-191.
Kavouras, Stavros A.; Johnson, Evan C.; Bougatsas, Dimitris; Arnaoutis, Giannis; Panagiotakos, Demosthenes B.; Perrier, Erica; Klein, Alexis (2016): Validation of a urine color scale for assessment of urine osmolality in healthy children. In: European journal of nutrition 55 (3), S. 907–915. DOI: 10.1007/s00394-015-0905-2.
Th. Gries, Stefan (2015): The most under-used statistical method in corpus linguistics: multi-level (and mixed-effects) models. In: Corpora 10 (1), S. 95–125. DOI: 10.3366/cor.2015.0068.
Reviewer 2 Report
This manuscript describes a study of the effects of water consumption on cognition in school children. It describes an intervention with respect to education on healthy drinking behaviour, but in reality there were no significant differences in fluid consumption between the two groups, suggesting that the intervention 'failed'. There were differences in self-reported thirst and urine colour between the two groups, which might suggest a 'Hawthorne Effect'.
The results a discussed fully and support the conclusion that adequate hydration improves cognition, but that there is a 'maximal effect', i.e. further hydration has no beneficial effect.
One minor query: Table 2, Corsi Block, LOngest Path: All groups have identical median values yet Kruskal-Wallis indicates a significant difference between groups. Can this occur due to interquartile range differences only?
Author Response
Response to Reviewer 2 Comments
We thank the reviewer for her/his thorough and careful evaluation of our paper and for the constructive critique, comments .and suggestions. We considered all issues addressed and revised our paper accordingly.
1. One minor query: Table 2, Corsi Block, LOngest Path: All groups have identical median values yet Kruskal-Wallis indicates a significant difference between groups. Can this occur due to interquartile range differences only?
Your conclusion is correct. The results are presented as median, 25th and 75th quartiles because most of the values are not normally distributed. The mean values for this analysis differed between groups (group 1: 4.9 ± 0.8, group 2: 5.2 ± 0.8, group 3: 5.3 ± 0.8, group 4: 5.0 ± 0.8) and could give the reader a better impression. However, the mean average representation is not recommended for not normally distributed results. For the cognitive parameter Longest Path, we observed low scatter and high results within the groups. The children achieved an average of 5 for the possible 6 points. Furthermore, the Kruskal Wallis test is not a median test but a rank sum test. It ranks all of the observations and then sums the ranks from both groups. Thus, it is possible for groups to have different rank sums and yet have equal medians.
Reviewer 3 Report
Thank you for the opportunity to review this manuscript. I believe the study is well conducted and the paper is well written. I have a few minor points of questions:
1) It is unclear as to how exactly water consumption was recorded. Was there visual inspection of the water bottles and the levels were recorded? Were there measurements written on the bottles themselves?
2) The authors state that there were no TWI advantages over 1.5L. Does that suggest an "optimal" water consumption point and when that threshold is crossed, there are no longer advantages? I suggest the authors address this a little more in the discussion
3) The authors state that the intervention group reported higher levels of subjective thirst at midday. Why would this be? The authors have addressed this a little bit in the discussion, but I feel more attention and works cited are needed to clarify this point
Author Response
Response to Reviewer 3 Comments
We thank the reviewer for her/his thorough and careful evaluation of our paper and for the constructive critique, comments .and suggestions. We considered all issues addressed and revised our paper accordingly.
1. It is unclear as to how exactly water consumption was recorded. Was there visual inspection of the water bottles and the levels were recorded? Were there measurements written on the bottles themselves?
This is an important note. For the intervention, we used scaled bottles that were initially filled up to 500 ml and were refilled as soon as they were completely empty. At the end of the intervention and before the cognitive tests, the bottles were collected and weighed using a kitchen scale. The bottle weight was then subtracted. For a better understanding, we have completed the information on the weighing protocol (page 4, line 13-20).
2. The authors state that there were no TWI advantages over 1.5L. Does that suggest an "optimal" water consumption point and when that threshold is crossed, there are no longer advantages? I suggest the authors address this a little more in the discussion.
We agree that we haven't paid as much attention to this aspect. As explained in the discussion (page 15, line 21-28), few studies only have examined these mechanisms in children. To our knowledge, there is no study that looked at hyperhydration in healthy children in this context. That is why we have only drawn cautious conclusions. Due to the lack of literature, we decided to keep the comments short.
3. The authors state that the intervention group reported higher levels of subjective thirst at midday. Why would this be? The authors have addressed this a little bit in the discussion, but I feel more attention and works cited are needed to clarify this point
Similar to point 2, studies with healthy children in this context are rare and therefore the thirst mechanisms in children cannot be fully explained. In order to interpret the observation mentioned, we considered a psychological aspect - the Hawthorne effect (page 16, line 19-24). The Hawthorne effect concerns research participation, the consequent awareness of being studied, and possible impact on behavior (McCambridge et al., 2014). This could, in individual cases, lead to an under- or overestimate of the self-reported thirst.
References
McCambridge, Jim; Witton, John; Elbourne, Diana R. (2014): Systematic review of the Hawthorne effect: new concepts are needed to study research participation effects. In: Journal of clinical epidemiology 67 (3), S. 267–277. DOI: 10.1016/j.jclinepi.2013.08.015.